# Sex differences in alcohol and tobacco use in Ugandan adults: A log-linear analysis of interaction patterns

Grace Kakaire[1]*, Edna Chepkemoi Chumoh[2]

1 Department of Statistical Methods and Actuarial Sciences, School of Statistics and Planning, Makerere University, Kampala, Uganda, 2 Department of Mathematics, Physics and Computing, School of Science and Aerospace Studies, Moi University, Eldoret, Kenya

* kakairegrace2@gmail.com

## Abstract

Alcohol and tobacco use remain major contributors to non-communicable disease burden worldwide. Understanding how these behaviors intersect across sexes is crucial for effective prevention strategies. This study investigated the interrelationships between sex, alcohol use, and tobacco use among Ugandan adults using log-linear modeling. Data were drawn from the 2014 Uganda WHO STEPwise Approach to Non-Communicable Disease Risk Factor Surveillance (STEPS) survey, a nationally representative cross-sectional study including 3,987 adults aged 18–69 years. "Current use" was defined as self-reported consumption of alcohol or tobacco within the past 30 days. Frequencies and proportions were stratified by sex, and a series of hierarchical log-linear models were fitted to assess independence and interaction effects. Model fit was evaluated using the Likelihood-Ratio Chi-square ($G^2$) and Dissimilarity Index (D) statistics. Alcohol use was more prevalent among females (60.0%, 95% CI: 56.8–63.1%) than males (38.9%, 95% CI: 36.2–41.7%), while tobacco use was slightly higher among males (9.05%, 95% CI: 7.4–10.9%) than females (7.95%, 95% CI: 6.4–9.8%). Combined alcohol and tobacco use was relatively uncommon but marginally higher in females (4.89%, 95% CI: 3.7–6.4%) than males (3.20%, 95% CI: 2.3–4.5%). The Homogeneous Association model provided the best fit ($G^2 = 0.93$, $p = 0.334$, $D = 0.5201$), indicating that two-way interactions adequately captured the observed data. The Alcohol × Tobacco interaction was non-significant ($p = 0.50$), suggesting that alcohol and tobacco behaviors occurred independently once sex differences were accounted for—implying distinct behavioral or social drivers for each substance. This analysis revealed sex-specific patterns in substance use, with higher alcohol consumption among women and greater tobacco use among men. The absence of a significant alcohol–tobacco interaction suggests these behaviors operate within different social or environmental contexts rather than as co-occurring habits. These findings underscore the importance of sex-sensitive

**Data availability statement:** The data underlying the results presented in the study are available from https://figshare.com/s/e817a72eb0d1f72f78a4.

**Funding:** The author(s) received no specific funding for this work.

and behavior-specific prevention strategies and highlight the utility of log-linear modeling for exploring complex behavioral associations in public health data.

## Introduction

Substance use, particularly alcohol and tobacco consumption, poses a significant public health concern globally. These behaviors are not only among the leading causes of preventable morbidity and mortality but also present substantial socio-economic burdens on healthcare systems and societies at large [1,2]. Understanding the patterns and interrelations of substance use is essential for developing effective prevention strategies, targeted interventions, and policy responses [3]. While alcohol and tobacco are often studied separately, growing evidence suggests that their usage is frequently interrelated, with overlapping risk factors, behavioral motivations, and demographic distributions [4–6].

Sex is a well-documented demographic variable that influences both the prevalence and patterns of alcohol and tobacco use. Traditionally, substance use has been more common among males; however, recent trends indicate a narrowing of the gender gap, particularly with respect to alcohol consumption [7–9]. Women are increasingly engaging in alcohol use, and in some populations, prevalence rates have surpassed those of men [9,10]. Conversely, tobacco use remains slightly more prevalent among men [5,7,11]. These shifting patterns warrant continuous investigation, particularly to uncover whether the associations between alcohol and tobacco use differ across sex and whether these behaviors interact or operate independently [7,8,12].

The use of log-linear models offers a powerful statistical framework for exploring such associations in categorical data. Unlike traditional regression models, which often assume a dependent outcome, log-linear analysis treats all variables symmetrically, making it particularly suitable for examining interactions among categorical variables without implying causality [13–19]. This methodological strength allows for the assessment of mutual, joint, and conditional independence, as well as higher-order interaction effects [5,7,14,20].

This study aims to explore the interrelationship between sex, alcohol use, and tobacco use in a population-based sample using log-linear modelling. The objectives are threefold: (1) to describe the prevalence of alcohol, tobacco, and combined use stratified by sex; (2) to evaluate different log-linear models to identify the best-fitting representation of the data; and (3) to interpret the patterns of interaction among the variables to provide insights into substance use behaviors. The study incorporates model fit statistics such as the likelihood ratio chi-square ($G^2$) and the Dissimilarity Index (D), which offer robust criteria for model evaluation [5,7].

By integrating descriptive statistics with advanced categorical modelling techniques, this research contributes to a nuanced understanding of substance use dynamics across sex. The findings have implications for public health programming, particularly in designing interventions that are tailored to demographic subgroups and sensitive to the co-use or independence of substance behaviors [8].

## Materials and methods

This study employed a quantitative, cross-sectional design to explore the association between alcohol use, tobacco use, and their co-occurrence, while accounting for sex as a stratification and potential confounding variable.

### Study design and data source

The data used in this study were derived from the World Health Organization (WHO) STEPwise Approach to Non-Communicable Disease Risk Factor Surveillance (STEPS) survey conducted in Uganda. The WHO STEPS survey is a standardized, population-based tool designed to collect nationally representative data on major risk factors for non-communicable diseases (NCDs), including tobacco use, alcohol consumption, physical inactivity, diet, and metabolic measures such as blood pressure and body mass index. The survey employs a multistage cluster sampling design to ensure representativeness across geographic and demographic strata. Data are collected in three sequential "steps": Step 1 gathers information on behavioral risk factors using a structured questionnaire; Step 2 involves physical measurements such as height, weight, and blood pressure; and Step 3 includes biochemical assessments (e.g., glucose and cholesterol levels). For this study, only Step 1 data on behavioral risk factors—specifically alcohol and tobacco use, were utilized.

### Study setting and population

The data were obtained from the nationally representative Uganda WHO STEPS Non-Communicable Disease Risk Factor Survey, conducted across all major administrative regions of the country. The survey employed a multistage cluster sampling design to capture both urban and rural populations, ensuring representativeness at the national level. Enumeration areas were selected proportionally to population size within each stratum, and households were randomly sampled within these areas. Eligible participants were adults aged 18 years and older residing in the selected households. Data collection followed WHO's standardized STEPS protocol, which includes detailed procedures for sampling, training, and quality assurance to ensure comparability across settings.

### Ethical statement

The 2014 Uganda STEPS NCD dataset is publicly available from the WHO NCD Microdata Repository. Ethical clearance was previously obtained from the Saint Francis Hospital, Nsambya Institutional Review Board (2006; renewed 2013). All data analyzed were de-identified, and no further ethical approval was required for secondary analysis.

### Variables and measurements

**Variables of Interest.** The primary variables are; Alcohol Use: Coded as "Yes" (if reported current alcohol use) or "No" and Tobacco Use: Coded as "Yes" (if reported current tobacco use in any form) or "No". The stratification Variable was sex; coded as Male or Female.

**Model specification.** In the log-linear framework, all variables are treated as categorical factors within a joint probability model, and no single variable is designated as dependent or independent. Instead, the model examines how the combination of variable levels contributes to the overall cell frequencies in the contingency table. In this study, *Sex* was included as a **stratification (or confounding) variable**, while *Alcohol use* and *Tobacco use* were considered the primary behavioral factors of interest. This approach allows for the assessment of whether the joint distribution of alcohol and tobacco use differs systematically by sex, without assuming a causal direction among variables.

**Model scope and variable selection.** The study focused on three key categorical variables; Sex, Alcohol use, and Tobacco use, to examine their interrelationships using log-linear modeling. While other variables such as age, education, or socioeconomic status may also influence substance use, these were excluded to maintain model parsimony and prevent sparse cell counts, which can compromise model stability and convergence in high-dimensional

contingency tables. The log-linear approach emphasizes understanding the structure of associations rather than causal inference; therefore, limiting the model to the most conceptually relevant variables was statistically and substantively justified.

**Role of sex in analysis.** In this study, sex was incorporated at two analytical stages. First, during descriptive analysis, stratification by sex was performed to compare the prevalence of alcohol and tobacco use separately among males and females. Second, in the log-linear modelling, sex was retained as an active variable and allowed to interact with both alcohol and tobacco use (i.e., Sex×Alcohol and Sex×Tobacco terms). This approach enabled evaluation of whether the joint distribution of alcohol and tobacco use varied significantly by sex, while also accounting for potential confounding effects. Thus, sex served both as a stratification factor in descriptive statistics and as an interaction term in the modelling framework.

## Statistical analysis

Data cleaning and statistical analysis were conducted using R version 4.4.3 with support from the haven, dplyr, ggplot2, MASS, and cowplot packages.

Categorical variables for alcohol use, tobacco use, and sex were converted into factors and relabeled as "Alcohol", "Tobacco", and "Sex", respectively. A contingency table was created to summarize the frequency of all possible combinations of these variables. Observations with missing data were excluded from all analyses.

## Stratification by sex

In this study, sex was treated as a stratification and potential confounding variable rather than a primary exposure. All prevalence estimates and model interpretations are accordingly stratified or adjusted for sex. Analyses were stratified by sex to allow for comparison of prevalence and associations within each sex category (i.e., male and female). This approach helps control for potential confounding effects of sex on the observed relationships between alcohol and tobacco use.

## Sample size and prevalence estimation

The data used in this analysis were obtained from the World Health Organization's 2014 Uganda STEPS survey, which employed multistage cluster sampling to produce nationally representative estimates of non-communicable disease risk factors among adults aged 18–69 years. The total survey included N = 3,987 respondents, of which all had complete information on sex, alcohol use, and tobacco use and were included in the present analysis.

Prevalence estimates were calculated by dividing the frequency of "Yes" responses for each behaviour by the total number of respondents in each sex category, expressed as a percentage. This approach ensured comparability of prevalence across sexes and behaviors and aligns with standard epidemiological reporting practices. Percentages presented in Table 1 were computed as follows:

**Table 1. Prevalence of Alcohol, Tobacco, and Combined Use by Sex.**

| Sex | Combination | Frequency | Percentage |
|---|---|---|---|
| Female | Alcohol yes | 920 | 60.0 |
| Male | Alcohol yes | 912 | 38.9 |
| Female | Tobacco yes | 122 | 7.95 |
| Male | Tobacco yes | 212 | 9.05 |
| Female | Combo yes yes | 75 | 4.89 |
| Male | Combo yes yes | 75 | 3.20 |

$$\text{Percentage} = \frac{\text{Frequency of each combination (Yes)}}{\text{Total number of respondents by sex}} \times 100$$

These results were summarized in a table and visualized using a grouped bar chart.

## Log-linear modeling

To investigate associations among the three categorical variables—Sex, Alcohol, and Tobacco—a series of log-linear models were fitted using the loglm() function from the MASS package in R. These models describe the natural logarithm of expected cell counts in the contingency table as linear combinations of main effects and interactions.

## Rationale for log-linear modelling

The log-linear model was employed to explore associations among categorical variables; Sex, Alcohol use, and Tobacco use, without imposing a directional relationship. This approach is particularly suitable when the goal is to examine the structure of associations rather than predict a specific outcome. By modeling the natural logarithm of expected cell counts as a function of main effects and interactions, log-linear analysis captures how the joint distribution of variables deviates from independence. Unlike logistic regression, which specifies a single dependent variable, the log-linear model treats all factors symmetrically, allowing for the detection of mutual, joint, and conditional independence and higher-order interaction effects. This makes it a powerful tool for identifying complex interdependencies in categorical data. Additionally, hierarchical log-linear models permit formal statistical comparison using likelihood ratio chi-square ($G^2$) and Dissimilarity Index (D) statistics, providing an objective framework for evaluating model fit and selecting the most parsimonious representation of the data.

Each model assumes different independence structures and is specified as follows:

## Independence model

All variables are mutually independent.

Model Equation:

$$\log(\mu_{ijk}) = \lambda + \lambda_i^{\text{Sex}} + \lambda_j^{\text{Alcohol}} + \lambda_k^{\text{Tobacco}} \tag{1}$$

where $\mu_{ijk}$ is the expected count for Sex = $i$, Alcohol = $j$, and Tobacco = $k$.

## Joint independence model

Sex is independent of the interaction between Alcohol and Tobacco.

Model Equation:

$$\log(\mu_{ijk}) = \lambda + \lambda_i^{\text{Sex}} + \lambda_{jk}^{\text{Alcohol} \times \text{Tobacco}} \tag{2}$$

## Conditional independence model

Allows interactions between Sex and each exposure separately but assumes Alcohol and Tobacco are conditionally independent given Sex.

Model Equation:

$$\log(\mu_{ijk}) = \lambda + \lambda_i^{\text{Sex}} + \lambda_j^{\text{Alcohol}} + \lambda_k^{\text{Tobacco}} + \lambda_{ij}^{\text{Sex} \times \text{Alcohol}} + \lambda_{ik}^{\text{Sex} \times \text{Tobacco}} \tag{3}$$

## Homogeneous association model

Includes all pairwise interactions but excludes the three-way interaction.
Model Equation:

$$\log(\mu_{ijk}) = \lambda + \lambda_{ij}^{Sex \times Alcohol} + \lambda_{ik}^{Sex \times Tobacco} + \lambda_{jk}^{Alcohol \times Tobacco} \qquad (4)$$

## Saturated model

Includes all main effects and all possible interactions, including the three-way interaction; perfectly fits the observed data.
Model Equation:

$$\log(\mu_{ijk}) = \lambda + \lambda_i^{Sex} + \lambda_j^{Alcohol} + \lambda_k^{Tobacco} + \lambda_{ij}^{Sex \times Alcohol} + \lambda_{ik}^{Sex \times Tobacco} + \lambda_{jk}^{Alcohol \times Tobacco} + \lambda_{ijk}^{Sex \times Alcohol \times Tobacco} \qquad (5)$$

Each of these models was also estimated using Poisson regression via the glm() function with the log link function and Poisson family. This allowed for the extraction of parameter estimates, standard errors, Z-values, and p-values, which aid in assessing the statistical significance of model terms.

## Rationale for model selection

The choice of the five hierarchical log-linear models—Mutual Independence, Joint Independence, Conditional Independence, Homogeneous Association, and Saturated—was guided by established theoretical frameworks in categorical data analysis [13,21]. These models represent successive levels of complexity in the relationships among three categorical variables. Beginning with the simplest assumption of complete independence, each subsequent model relaxes one or more independence constraints, allowing for the systematic assessment of pairwise and higher-order interactions. This approach aligns with prior applications in epidemiology and social science research, where the goal is to identify the most parsimonious model that explains observed associations without overfitting [22]. Thus, the inclusion of these specific five models is theoretically grounded and consistent with best practices in log-linear modelling.

## Model estimation and evaluation

Each model was also estimated using Poisson regression (glm()) with a log link function to obtain parameter estimates, standard errors, Z-values, and p-values. These values assess the significance of main effects and interactions.
To evaluate model fit, two metrics were used

## Likelihood-ratio chi-square (G² Statistic)

The $G^2$ statistic compares observed and expected cell counts under each model:

$$G^2 = 2 \sum O \cdot \log\left(\frac{O}{E}\right) \qquad (6)$$

Where: O = Observed frequency and E = Expected frequency under the model
A lower $G^2$ value indicates a better-fitting model. A non-significant $G^2$ ($p > 0.05$) suggests that the model fits the data well.

## Dissimilarity index (D)

The Dissimilarity Index (D) was used to measure the overall discrepancy between observed and expected values, calculated as:

$$D = \frac{\sum |O - E|}{2 \sum O}$$

(7)

Where O = Observed frequency and E = Expected frequency under the model.

This metric quantifies the proportion of observations that would need to be reallocated for the expected counts to match the observed. A lower D value indicates better fit, with D = 0 for the saturated model by definition.

### Justification for poisson regression (GLM Framework)

In addition to the log-linear modeling implemented through the loglm() function, the same models were estimated using Poisson regression within the Generalized Linear Model (GLM) framework. This approach is appropriate because the outcome of interest comprises cell counts derived from combinations of categorical variables (Sex, Alcohol use, and Tobacco use), which follow a Poisson distribution. The Poisson GLM with a log link function models the natural logarithm of expected counts as a linear function of the categorical predictors and their interactions. This formulation is mathematically equivalent to the log-linear model but offers additional flexibility for parameter estimation and inference.

Through the GLM approach, we obtained parameter estimates, standard errors, Z-values, and p-values, facilitating statistical testing of main and interaction effects. Moreover, this framework allows for straightforward comparison of nested models using likelihood ratio tests and provides familiar goodness-of-fit measures such as the deviance ($G^2$ statistic). Employing both the log-linear and Poisson GLM formulations therefore strengthens the robustness and interpretability of our findings.

## Results

### Prevalence of alcohol, tobacco, and combined use by sex

Figure 1 and Table 1 jointly present the prevalence of alcohol use, tobacco use, and their combination, stratified by sex. The bar chart visually highlights the sex-based disparities in substance use, while the accompanying table provides exact frequencies and percentages.

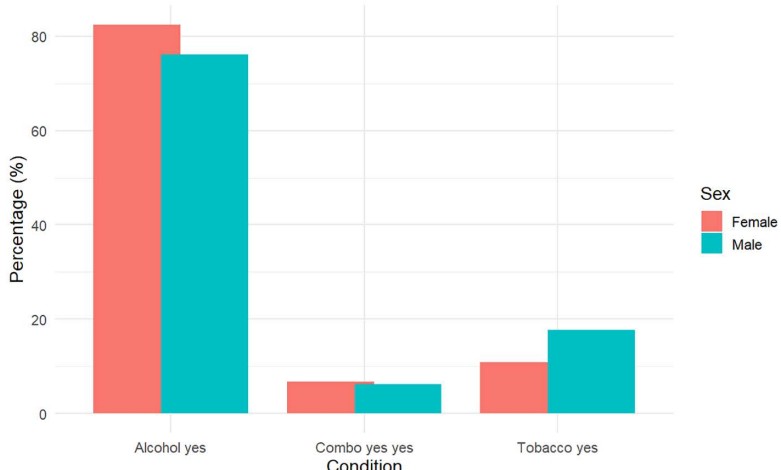

**Fig 1. Prevalence of Alcohol, Tobacco, and Combined Use by Sex.**

A clear difference is evident in alcohol use between females and males. The bar chart shows a higher bar for females under "Alcohol yes", which is corroborated by the table showing 60.0% of females reported alcohol use compared to 38.9% of males. This indicates that alcohol consumption is substantially more common among females in this sample, an unexpected pattern given general trends in substance use literature and worth further investigation.

In contrast, tobacco use was more prevalent among males. The chart reflects this with a taller bar for males in the "Tobacco yes" category, aligning with the table which shows 9.05% prevalence among males versus 7.95% among females. While the difference is modest, it is consistent with broader patterns seen in other studies, where tobacco use tends to be more common among men.

Both the graph and the table show that combined alcohol and tobacco use is relatively uncommon in both sexes. However, a slightly higher percentage of females (4.89%) reported using both substances compared to males (3.20%). This is visually confirmed by the taller "Combo yes yes" bar for females in the chart. This subtle sex difference in dual use may point toward different patterns of concurrent substance engagement.

Together, the chart and table provide a clear, cohesive view of how substance use behaviors differ by sex in this cohort. While alcohol use is significantly higher among females, tobacco and combined use are slightly more prevalent in males. The alignment between numerical data and graphical representation reinforces the reliability of these observations and supports the need to account for sex as a stratifying variable in further multivariable analyses; log-linear modeling.

## Log-linear model results

To evaluate the structure of associations among Sex, Alcohol Use, and Tobacco Use, a series of hierarchical log-linear models were fitted and assessed using the likelihood ratio statistic ($G^2$), degrees of freedom (df), associated p-values, and the Dissimilarity Index (D). Table 2 presents the results.

The Mutual Independence Model assumes that all three variables are independent. It performed poorly, as indicated by a large $G^2$ value of 168.19 (df = 4, p < 0.001) and the highest dissimilarity index (D = 0.5704), signaling a poor fit and substantial deviation between observed and expected counts.

The Joint Independence Model assumes sex is independent of the alcohol–tobacco interaction, this model also failed to fit the data adequately ($G^2$ = 167.38, df = 3, p < 0.001), with a marginally lower D (0.5684), suggesting little improvement.

With regard to the Conditional Independence Model, by allowing interaction between sex and each exposure individually (alcohol and tobacco), model fit dramatically improved ($G^2$ = 1.38, df = 2, p = 0.501), indicating no significant lack of fit. The Dissimilarity Index dropped to 0.5205, reflecting reduced discrepancy between observed and expected values.

The Homogeneous Association Model included all pairwise interactions but not the three-way interaction. It had the best overall fit among reduced models ($G^2$ = 0.93, df = 1, p = 0.334) and the lowest dissimilarity index (D = 0.5201), very close to that of the saturated model.

The Saturated Model as expected ($G^2$ = 0, df = 0, p = 1) perfectly fit the data, with a D of 0.5181, the lowest possible by definition. However, because the homogeneous association model provides a statistically equivalent fit with one fewer parameter, it is preferred for parsimony.

**Table 2. Log-Linear Model Fit Statistics Evaluating Associations Among Sex, Alcohol Use, and Tobacco Use.**

| Model | Formula | L-R (G²) | df | P-value | Dissimilarity Index (D) |
|---|---|---|---|---|---|
| Mutual Independence | ~Sex + Alcohol + Tobacco | 168.19 | 4 | <0.001 | 0.5704 |
| Joint Independence | ~Sex + Alcohol*Tobacco | 167.38 | 3 | <0.001 | 0.5684 |
| Conditional Independence | ~Sex + Alcohol + Tobacco + Sex*Alcohol + Sex*Tobacco | 1.38 | 2 | 0.501 | 0.5205 |
| Homogeneous Association | ~Sex*Alcohol + Sex*Tobacco + Alcohol*Tobacco | 0.93 | 1 | 0.334 | 0.5201 |
| Saturated | ~Sex * Alcohol * Tobacco | 0 | 0 | 1 | 0.5181 |

The poor fit of the independence models underscores the presence of associations among sex, alcohol use, and tobacco use. The conditional and homogeneous association models fit the data well, with the latter offering the best balance between goodness-of-fit and model simplicity. These results justify further interpretation of interaction effects in stratified or multivariable models.

## Observed vs. expected frequencies across log-linear models

Table 3 compares the observed counts with expected frequencies generated by five log-linear models: Mutual Independence, Joint Independence, Conditional Independence, Homogeneous Association, and Saturated. These expected values offer insight into how well each model captures the underlying joint distribution of sex, alcohol use, and tobacco use.

The Mutual and Joint Independence Models show substantial deviations from the observed counts. For example, in the "Male/ Yes Alcohol/ No Tobacco" group, the mutual independence model overestimates the frequency (1011.56 vs. 837 observed), and similarly for the joint independence model (1016.32).

Across most cells, both models either significantly overestimate or underestimate counts, especially in cells involving tobacco use. This is consistent with their poor goodness-of-fit statistics ($G^2 \approx 167$, $p < 0.001$).

The Conditional Independence Model assumes that alcohol and tobacco use are conditionally independent given sex. It provides much closer estimates for most cells. For instance, in the "Female/ Yes Alcohol/ No Tobacco" category, the expected count (846.83) closely approximates the observed (845). Minor discrepancies are still present, particularly in the "Male/ No Alcohol/ No Tobacco" group, but they are reduced compared to the simpler models.

With all pairwise interactions included in the Homogeneous Association Model, this model produces expected values that nearly mirror the observed data. The fit is particularly close in the "Male/ No Alcohol/ No Tobacco" (1296.97 vs. 1293 observed) and "Female/ No Alcohol/ Yes Tobacco" (50.97 vs. 47) groups. The minimal differences between observed and expected counts support the model's superior fit ($G^2 = 0.93$, $p = 0.334$), indicating that pairwise interactions sufficiently account for the associations without the need for a full three-way interaction.

Saturated Model as expected, replicates the observed frequencies exactly across all cells, since it includes all possible interaction terms. However, it does so at the cost of greater model complexity.

The pattern of discrepancies shows that independence assumptions (Mutual or Joint) fail to capture the structure of the data. The Homogeneous Association model strikes the best balance between simplicity and fit, with expected frequencies nearly identical to observed values. These results reinforce the presence of meaningful pairwise interactions among sex, alcohol use, and tobacco use, justifying the use of more complex models over those assuming independence.

The deviations between observed and expected cell counts across the log-linear models reveal important behavioral patterns in alcohol and tobacco use between males and females. Under the independence models, male alcohol use was substantially overestimated (expected ≈ 1010 vs. observed 837), while female alcohol use was underestimated

**Table 3. Observed and Expected Cell Counts from Log-Linear Models for the Joint Distribution of Sex, Alcohol Use, and Tobacco Use.**

| Sex | Alcohol | Tobacco | Observed | Independent | Joint | Conditional | Homogeneous | Saturated |
|---|---|---|---|---|---|---|---|---|
| Male | Yes | No | 837 | 1011.56 | 1016.32 | 829.44 | 833.03 | 837 |
| | | Yes | 75 | 95.39 | 90.63 | 82.56 | 78.97 | 75 |
| | No | No | 1293 | 1128.62 | 1123.87 | 1300.56 | 1296.97 | 1293 |
| | | Yes | 137 | 106.43 | 111.18 | 129.44 | 133.03 | 137 |
| Female | Yes | No | 845 | 662.57 | 665.68 | 846.83 | 848.97 | 845 |
| | | Yes | 75 | 62.48 | 59.37 | 73.17 | 71.03 | 75 |
| | No | No | 567 | 739.24 | 736.13 | 565.17 | 563.03 | 567 |
| | | Yes | 47 | 69.71 | 72.82 | 48.83 | 50.97 | 47 |

(expected ≈ 663 vs. observed 845), suggesting that alcohol consumption is notably higher among females than would be expected under independence. Conversely, tobacco use among males was underestimated (expected ≈ 106 vs. observed 137), indicating a stronger-than-expected tendency toward smoking in men. These systematic deviations diminish markedly under the Conditional and Homogeneous Association models, where expected counts align closely with observed data (e.g., *Male/ No Alcohol/ No Tobacco*: 1296 vs. 1293; *Female/ Alcohol Yes/ No Tobacco*: 849 vs. 845). This convergence demonstrates that accounting for pairwise interactions between sex and each behavior accurately captures the observed data structure. Overall, these patterns point to a behavioral divide; female-dominant alcohol use and male-dominant tobacco use; while co-use remains relatively infrequent and statistically independent once sex is considered.

### Parameter estimates from the parsimonious model

Table 4 presents the parameter estimates from the Homogeneous Association Model, which includes all pairwise interactions among the variables (Sex, Alcohol, and Tobacco) but excludes the three-way interaction. These estimates are on the log scale and reflect deviations from the baseline category for each interaction. The following are the key findings:

With regard to Sex × Alcohol Interaction, the interaction between Female and No Alcohol is significantly negative (Estimate = −0.3917, p < 0.001), indicating that females are less likely to be in the no-alcohol category compared to the reference. Conversely, Male with No Alcohol shows a positive and significant association (Estimate = 0.4427, p < 0.001), suggesting a higher likelihood for males to fall into the no-alcohol group. The Female × Alcohol Yes interaction is not significant (p = 0.69), implying no meaningful departure from the baseline for this group.

With regard to Sex × Tobacco Interaction, Both Female × Tobacco Yes (Estimate = −2.4021) and Male × Tobacco Yes (Estimate = −2.2772) are highly significant and negative, indicating a substantially lower likelihood of tobacco use among both sexes, with females slightly less likely than males to be in the tobacco-use category.

Alcohol × Tobacco Interaction, the Alcohol Yes × Tobacco Yes interaction is not significant (p = 0.50), suggesting no strong evidence of dependence between alcohol and tobacco use after accounting for sex interactions. This aligns with the model structure where sex-stratified relationships dominate.

The results confirm significant sex-specific patterns in both alcohol and tobacco use. While tobacco use is consistently low across both sexes, sex modifies alcohol use differently, especially in the no-alcohol group. The lack of a significant alcohol–tobacco interaction suggests these behaviors are not jointly patterned beyond what is explained by sex.

### Discussion

This study explored the interrelationships between sex, alcohol use, and tobacco use using log-linear modelling to examine associations and interaction effects among categorical variables [17–19,21–23]. The integrated interpretation of prevalence statistics, goodness-of-fit measures, and model parameter estimates provides a comprehensive understanding of the underlying structure of substance use behavior in this population [16].

**Table 4. Parameter Estimates from the Parsimonious Log-Linear Model (Homogeneous Association Model).**

| Parameter | Estimate | Std. Error | Z-value | P-value |
|---|---|---|---|---|
| SexFemale*AlcoholNo | −0.3917 | 0.0545 | −7.19 | < 0.001 |
| SexMale*AlcoholNo | 0.4427 | 0.0437 | 10.14 | < 0.001 |
| SexFemale*AlcoholYes | 0.0190 | 0.0478 | 0.40 | 0.69 |
| SexFemale*TobaccoYes | −2.4021 | 0.1167 | −20.59 | < 0.001 |
| SexMale*TobaccoYes | −2.2772 | 0.0844 | −26.99 | < 0.001 |
| AlcoholYes*TobaccoYes | −0.0788 | 0.1175 | −0.67 | 0.50 |

The descriptive analysis revealed notable sex differences in substance use patterns. Alcohol consumption was significantly more prevalent among females (60%) than males (38.9%), while tobacco use was slightly more common among males (9.05%) compared to females (7.95%) [24–26]. Combined use of alcohol and tobacco was relatively low but more pronounced among females (4.89%) than males (3.20%) [4,16,27]. These figures suggest that while alcohol use is more prominent among women, men may engage in tobacco use slightly more frequently [9,24,25,28]. The unexpectedly higher prevalence of alcohol use among females observed in this study may reflect a combination of sociocultural, psychosocial, and biological factors. Recent shifts in gender norms and increasing social acceptance of female drinking in Uganda and similar contexts have contributed to narrowing gender gaps in alcohol use [2,3]. Targeted alcohol marketing campaigns, particularly those emphasizing "modern" or "empowered" femininity, may also play a role. Additionally, women may experience unique psychosocial stressors—such as caregiving responsibilities, gender-based violence, or economic hardship—that increase vulnerability to alcohol use as a coping mechanism [6,8]. Biological differences in alcohol metabolism, including lower gastric alcohol dehydrogenase activity and higher blood alcohol concentrations per unit of intake, may further reinforce consumption patterns through faster onset of psychoactive effects [10,28]. Together, these factors highlight the complex interplay between gender, biology, and sociocultural change in shaping contemporary substance use patterns.

Log-linear model comparisons revealed that the Mutual and Joint Independence models poorly fit the observed data, as indicated by high likelihood ratio chi-square ($G^2$) values (168.19 and 167.38, respectively) and large dissimilarity indices ($D > 0.56$), suggesting substantial deviation between the observed and expected counts [15]. Conversely, both the Conditional Independence and Homogeneous Association models demonstrated strong fit, with low $G^2$ values (1.38 and 0.93) and non-significant p-values ($p = 0.501$ and $p = 0.334$, respectively), indicating that including two-way interactions between sex, alcohol use, and tobacco use sufficiently captured the observed patterns without requiring the full complexity of the saturated model [22].

Parameter estimates from the selected Homogeneous Association model underscored critical interaction effects. Males were more likely to be non-drinkers than females [12,28], with a significant positive estimate for the SexMaleAlcoholNo interaction [29]. Interestingly, no significant difference was found between sexes in the AlcoholYes category, suggesting that gender differences in alcohol abstinence are more pronounced than in alcohol use per se [30–32]. Both sexes showed strong, significant negative associations with tobacco use, though the effect was more pronounced for females [12], indicating lower tobacco use prevalence [9,25,26]. Notably, the AlcoholYesTobaccoYes interaction was non-significant, suggesting that alcohol and tobacco use do not co-occur [8] more frequently than would be expected by their marginal distributions, once sex is accounted for [33,34].

The absence of a significant alcohol–tobacco interaction suggests that the two behaviours may be shaped by distinct social and environmental influences rather than occurring within the same behavioural contexts. In many low- and middle-income settings, including Uganda, alcohol consumption often occurs in social or celebratory contexts, frequently among women in household or community gatherings, while tobacco use; especially smoking, tends to be more individualistic, socially restricted, and predominantly male-dominated [29,30]. This divergence in social acceptability and context may explain why alcohol and tobacco use do not co-occur beyond what is predicted by sex differences alone. Moreover, policy enforcement and cultural norms often stigmatize female tobacco use more than alcohol consumption, reinforcing the behavioral separation between these substances. Thus, the lack of statistical interaction in this study likely reflects the existence of distinct psychosocial and normative pathways driving alcohol and tobacco use behaviors in Ugandan adults.

These findings reinforce the importance of considering sex as a moderating factor in substance use behaviors. The observed gender disparities in alcohol and tobacco use are consistent with broader epidemiological trends and may reflect sociocultural, psychological, or policy-driven influences [9,21,35,36]. The absence of a significant alcohol-tobacco interaction effect [8] implies that integrated prevention strategies should be sex-sensitive but not necessarily predicated on co-use patterns [21,28].

This study benefits from the application of log-linear modelling, which enables a robust analysis of multidimensional categorical data without assuming a causal direction [15–19,21]. The use of the $G^2$ statistic and Dissimilarity Index provided objective criteria for model selection and evaluation [22,30,31,34,37]. However, limitations include reliance on self-reported data, which may be biased, and a cross-sectional design, which precludes causal inference [5,38]. Additionally, categorical simplification may obscure more nuanced behavioral differences [20].

These results emphasize the need for targeted interventions and further research into sex-specific determinants of substance use. Longitudinal designs and inclusion of additional psychosocial variables could offer deeper insight into the causal mechanisms and help shape more effective public health strategies [9,22].

## Conclusion

This study employed log-linear modelling to investigate the associations between sex, alcohol use, and tobacco use within a categorical data framework. By fitting and comparing a series of nested models, we systematically examined the independence and interaction effects of these variables. The findings underscore the nuanced interplay between sex and substance use behaviors, providing important implications for public health surveillance and intervention design.

Descriptive results revealed that alcohol consumption was notably more prevalent among females, while tobacco use was marginally higher among males. The co-use of alcohol and tobacco was relatively uncommon overall but showed a slightly higher prevalence among women. These patterns suggest potential differences in social norms, health behaviors, or access to substances between the sexes.

The log-linear modelling approach allowed for an objective evaluation of the data structure. The Mutual and Joint Independence models failed to adequately fit the observed data, suggesting that simple additive relationships do not capture the complexities of substance use behavior. In contrast, the Conditional Independence and Homogeneous Association models, which incorporate key pairwise interactions, demonstrated strong fits without overfitting, as shown by non-significant $G^2$ values and lower dissimilarity indices. These models provided sufficient explanatory power without necessitating the fully saturated model.

Importantly, parameter estimates from the most parsimonious model (Homogeneous Association) highlighted meaningful sex-specific interactions. Males had a higher likelihood of being alcohol abstainers compared to females, and both sexes were significantly less likely to use tobacco, with the effect being stronger in females. The lack of a significant interaction between alcohol and tobacco use suggests that, once sex is accounted for, co-use does not exceed what would be expected by chance.

These findings have several practical implications. First, they advocate for sex-sensitive substance use prevention and intervention programs that recognize the distinct patterns and predictors of use in males and females. Second, the absence of a strong co-use effect between alcohol and tobacco, when stratified by sex, suggests that joint prevention strategies may not be necessary for all groups and should be tailored accordingly. Lastly, the study demonstrates the utility of log-linear models in public health research, especially when analyzing multi-way categorical data where traditional regression approaches may be less suitable.

This study contributes to the growing body of evidence on substance use disparities and highlights the value of advanced statistical modelling in public health research. Understanding how sex interacts with behavioral risk factors is essential for designing equitable and effective health interventions. The findings call for continued attention to gender-responsive health policy and suggest new avenues for targeted research in substance use epidemiology.

## Limitations

A key limitation of this study is the omission of additional sociodemographic and behavioral variables such as age, education, and income, which may also shape patterns of alcohol and tobacco use. Their exclusion was necessary to preserve the integrity of the log-linear framework, as inclusion of multiple categorical factors could have resulted in sparse

contingency cells and unreliable parameter estimates. However, this simplification limits the ability to fully capture confounding and interaction effects beyond sex. Future research employing larger datasets or alternative modeling frameworks (e.g., multinomial or structural equation modeling) should incorporate these variables to enhance explanatory power and generalizability.

Another key limitation of this study lies in the use of data from the 2014 WHO STEPS survey. Substance use behaviors are dynamic and can shift considerably over time due to changes in social norms, economic conditions, public health policies, and product availability. Over the past decade, Uganda has experienced significant socioeconomic development, urbanization, and shifts in lifestyle factors that may have altered patterns of alcohol and tobacco consumption. Additionally, the introduction of stricter tobacco control policies and evolving marketing and accessibility of alcoholic beverages could have influenced current use patterns. Consequently, the findings of this study should be interpreted as reflecting the behavioral landscape of the mid-2010s rather than current prevalence rates. Future research using updated STEPS data or longitudinal surveys is essential to capture emerging trends and assess whether these associations have persisted or evolved over time.

## Acknowledgments

The authors would like to thank the Uganda Ministry of Health, Non-Communicable Diseases (NCDs) Desk for unconditionally supporting this study by granting data access and providing administrative clearance.

## Author contributions

**Conceptualization:** Grace Kakaire.

**Data curation:** Grace Kakaire.

**Formal analysis:** Grace Kakaire.

**Investigation:** Grace Kakaire.

**Resources:** Grace Kakaire.

**Software:** Grace Kakaire.

**Validation:** Grace Kakaire, Edna Chepkemoi Chumoh.

**Visualization:** Grace Kakaire.

**Writing – original draft:** Grace Kakaire.

**Writing – review & editing:** Grace Kakaire, Edna Chepkemoi Chumoh.

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
