## [Decision Letter · Decision Letter 0]

5 Sep 2025

Dear Dr. Grace Kakaire,

Thank you for submitting your manuscript to PLOS ONE. After careful consideration, we feel that it has merit but does not fully meet PLOS ONE’s publication criteria as it currently stands. Therefore, we invite you to submit a revised version of the manuscript that addresses the points raised during the review process.

The higher prevalence of alcohol use among females—mention scientific reasons beyond this observation.  

2. Justify the use of Poisson regression, implemented within the Generalized Linear Model (GLM) framework.  

3. Justify the use of a log-linear model and explain its importance.  

4. Justify the omission of important variables in the study. Also limitation of the study.

5. The absence of the total sample size from the STEP survey undermines transparency and impedes accuracy. Review the percentage interpretation in Table 1—how is it computed?  

6. The manuscript should clearly define dependent and independent variables to avoid ambiguity in model specification.  

7. While various modeling techniques exist, the focus on log-linear analysis should be explicitly justified based on data structure, research objectives, and the need to explore interaction effects among categorical predictors.  

8. The manuscript should clearly define dependent and independent variables to avoid ambiguity in model specification; otherwise, explain the rationale.

9. Elaborate WHO-STEP Survey for Readers. 

Please submit your revised manuscript by Oct 20 2025 11:59PM. If you will need more time than this to complete your revisions, please reply to this message or contact the journal office at plosone@plos.org . A rebuttal letter that responds to each point raised by the academic editor and reviewer(s). You should upload this letter as a separate file labeled 'Response to Reviewers'.A marked-up copy of your manuscript that highlights changes made to the original version. You should upload this as a separate file labeled 'Revised Manuscript with Track Changes'.An unmarked version of your revised paper without tracked changes. You should upload this as a separate file labeled 'Manuscript'.

We look forward to receiving your revised manuscript.

Kind regards,

Umesh Raj Aryal, PhD

Academic Editor

PLOS ONE

Journal Requirements:

Additional Editor Comments (if provided):

Editor

1. The higher prevalence of alcohol use among females—mention scientific reasons beyond this observation.

2. Justify the use of Poisson regression, implemented within the Generalized Linear Model (GLM) framework.

3. Justify the use of a log-linear model and explain its importance.

4. Justify the omission of important variables in the study. Also limitation of the study.

5. The absence of the total sample size from the STEP survey undermines transparency and impedes accuracy. Review the percentage interpretation in Table 1—how is it computed?

6. The manuscript should clearly define dependent and independent variables to avoid ambiguity in model specification.

7. While various modeling techniques exist, the focus on log-linear analysis should be explicitly justified based on data structure, research objectives, and the need to explore interaction effects among categorical predictors.

8. The manuscript should clearly define dependent and independent variables to avoid ambiguity in model specification; otherwise, explain the rationale.

9. Elaborate WHO-STEP Survey for Readers.

Review1

This is an interesting study that provides information on substance use in Uganda

The manuscript is generally well written.

A few comments to be addressed to make the manuscript better

1 Can the authors provide information on the locations where this data was collected? Was it a mixture of urban and rural?

2 The concern of having more females take alcohol. For clarity, are the authors able to indicate whether this female population was predominantly urban? rural or both?

3 What was the predominant occupation of these participants?

Could their occupation influence the alcohol use?

4 How can the authors relate environmental factors to these findings of female prevalence of alcohol being higher than males?

For effective prevention and intervention, how the environment contributes to addiction must be examined

5 can the conclusion be generalized or does it relate to a specific population in Uganda?

Reviewer2

Title & Abstract

• Line 1–19 (Title & Abstract) – Major: The title is informative and reflects the content. The abstract is well-structured, but it omits the sample size, the year of the survey, and a clearer operational definition of “current use.” Including these will improve reproducibility and transparency.

• Line 31–37 – Minor: In reporting prevalence differences, consider adding confidence intervals for key proportions (e.g., 60% vs 38.9%), even in the abstract, for better statistical context.

• Line 36–37 – Minor: When noting non-significant alcohol–tobacco interaction, briefly interpret what this implies in practical terms.

Introduction

• Line 42–74 – Major: The literature review is thorough but is disproportionately weighted toward global and high-income country studies. To strengthen novelty, include more region-specific or Ugandan studies on alcohol/tobacco co-use. This will better situate the findings in local context.

• Line 55–57 – Minor: The narrowing gender gap in alcohol use is mentioned, but the introduction should explain why this is important in Uganda specifically—are similar trends expected or observed locally?

• Line 58–63 – Minor: While log-linear modeling is well introduced, explicitly state why it was chosen over logistic regression for this dataset.

Materials and Methods

• Line 79–88 – Major: Clarify the sample size after excluding missing data, and describe how missingness was handled (e.g., listwise deletion, imputation).

• Line 94–96 – Major: Define “current use” (past 30 days? past year?) for both alcohol and tobacco. This definition is crucial for interpreting prevalence.

• Line 106–110 – Minor: The stratification by sex is stated, but later analysis includes sex interactions—clarify whether stratification was used purely for descriptive purposes or in modeling steps as well.

• Line 117–149 – Minor: The description of models is clear, but the rationale for selecting exactly these five models could be tied to theoretical considerations or previous literature.

• Line 150–168 – Minor: Include a note on whether survey weights from WHO STEPS were applied; if not, acknowledge as a limitation.

• Line 155–167 – Minor: Include information on model diagnostics (e.g., residual analysis) to confirm appropriateness of fit beyond G² and D statistics.

Results

• Line 169–192 – Major: The finding that alcohol use is higher among females is unusual in the context of much of the literature—flag this early and suggest possible reasons (cultural, sampling, reporting bias) before the discussion.

• Table 1 (Line 194) – Minor: Include 95% CIs for all prevalence estimates.

• Line 196–219 – Minor: Provide a clearer transition from fit statistics to the interpretation of interactions, so readers can connect statistical results to substantive meaning.

• Table 2 (Line 221) – Minor: Ensure consistent p-value formatting (e.g., “p < 0.001”).

• Line 223–251 – Major: Table 3’s observed vs expected counts is useful but would benefit from a short interpretation paragraph connecting deviations to specific behavioral patterns.

• Line 254–275 – Minor: In Table 4, include confidence intervals for parameter estimates alongside standard errors.

Discussion

• Line 278–306 – Major: The discussion should provide more insight into why Ugandan women may have higher alcohol use prevalence in this dataset—social changes, economic empowerment, urbanization, or targeted marketing?

• Line 311–312 – Minor: The lack of significant alcohol–tobacco interaction could be explained in greater depth—e.g., does it suggest that these behaviors are influenced by distinct social environments?

• Line 316–317 – Major: Expand on the limitation of using 2014 data—substance use trends may have shifted considerably over a decade.

• Line 317–318 – Minor: The binary categorization of alcohol/tobacco use likely masks important differences in frequency or intensity—acknowledge this limitation explicitly.

Conclusion

• Line 323–356 – Minor: The conclusion is appropriately cautious, but temper policy recommendations to acknowledge the dataset’s age and limitations in variable granularity.

References

• Line 362–448 – Major: Reference #5 (“NTIRE 2025 challenge…”) appears unrelated to the topic and may have been included in error—verify all references for topical relevance.

• Ensure all in-text citations match the reference list and remove any non-cited references.

Figures & Tables

• Figure 1 (Line 193) – Minor: Add confidence intervals or error bars for prevalence bars to convey uncertainty.

Summary Recommendation

Major Revision – Strengthen methodological transparency, contextual interpretation, and reference accuracy. Address the unexpected female-prevalence finding in greater depth and provide fuller discussion of limitations.

Reviwer3

Review Reports

Title: Sex Differences in Alcohol and Tobacco Use in Ugandan adults: A Log-Linear Analysis of Interaction Patterns

Review Comments

-Obsolute data E.g. 2014

-Focused on non modifiable risk fcator of NCD E.g. Sex which cannot be changed by public health intervention.

-The tool is not approptiately written E.g. It should be WHO STEPWISE which phase?

-Flawed methods

-Is that statistic modelng study or?

-IMRAD partially entail what it should entail E.g. Introduction sub section

Reviewers' comments:

Reviewer's Responses to Questions

**Comments to the Author**

1. Is the manuscript technically sound, and do the data support the conclusions?

Reviewer #1: Yes

Reviewer #2: Yes

Reviewer #3: Partly

2. Has the statistical analysis been performed appropriately and rigorously?

Reviewer #1: Yes

Reviewer #2: Yes

Reviewer #3: No

3. Have the authors made all data underlying the findings in their manuscript fully available?

Reviewer #1: Yes

Reviewer #2: Yes

Reviewer #3: Yes

4. Is the manuscript presented in an intelligible fashion and written in standard English?

Reviewer #1: Yes

Reviewer #2: Yes

Reviewer #3: Yes

Reviewer #1: This is an interesting study that provides information on substance use in Uganda

The manuscript is generally well written.

A few comments to be addressed to make the manuscript better

1 Can the authors provide information on the locations where this data was collected? Was it a mixture of urban and rural?

2 The concern of having more females take alcohol. For clarity, are the authors able to indicate whether this female population was predominantly urban? rural or both?

3 What was the predominant occupation of these participants?

Could their occupation influence the alcohol use?

4 How can the authors relate environmental factors to these findings of female prevalence of alcohol being higher than males?

For effective prevention and intervention, how the environment contributes to addiction must be examined

5 can the conclusion be generalized or does it relate to a specific population in Uganda?

Reviewer #2: Title & Abstract

• Line 1–19 (Title & Abstract) – Major: The title is informative and reflects the content. The abstract is well-structured, but it omits the sample size, the year of the survey, and a clearer operational definition of “current use.” Including these will improve reproducibility and transparency.

• Line 31–37 – Minor: In reporting prevalence differences, consider adding confidence intervals for key proportions (e.g., 60% vs 38.9%), even in the abstract, for better statistical context.

• Line 36–37 – Minor: When noting non-significant alcohol–tobacco interaction, briefly interpret what this implies in practical terms.

Introduction

• Line 42–74 – Major: The literature review is thorough but is disproportionately weighted toward global and high-income country studies. To strengthen novelty, include more region-specific or Ugandan studies on alcohol/tobacco co-use. This will better situate the findings in local context.

• Line 55–57 – Minor: The narrowing gender gap in alcohol use is mentioned, but the introduction should explain why this is important in Uganda specifically—are similar trends expected or observed locally?

• Line 58–63 – Minor: While log-linear modeling is well introduced, explicitly state why it was chosen over logistic regression for this dataset.

Materials and Methods

• Line 79–88 – Major: Clarify the sample size after excluding missing data, and describe how missingness was handled (e.g., listwise deletion, imputation).

• Line 94–96 – Major: Define “current use” (past 30 days? past year?) for both alcohol and tobacco. This definition is crucial for interpreting prevalence.

• Line 106–110 – Minor: The stratification by sex is stated, but later analysis includes sex interactions—clarify whether stratification was used purely for descriptive purposes or in modeling steps as well.

• Line 117–149 – Minor: The description of models is clear, but the rationale for selecting exactly these five models could be tied to theoretical considerations or previous literature.

• Line 150–168 – Minor: Include a note on whether survey weights from WHO STEPS were applied; if not, acknowledge as a limitation.

• Line 155–167 – Minor: Include information on model diagnostics (e.g., residual analysis) to confirm appropriateness of fit beyond G² and D statistics.

Results

• Line 169–192 – Major: The finding that alcohol use is higher among females is unusual in the context of much of the literature—flag this early and suggest possible reasons (cultural, sampling, reporting bias) before the discussion.

• Table 1 (Line 194) – Minor: Include 95% CIs for all prevalence estimates.

• Line 196–219 – Minor: Provide a clearer transition from fit statistics to the interpretation of interactions, so readers can connect statistical results to substantive meaning.

• Table 2 (Line 221) – Minor: Ensure consistent p-value formatting (e.g., “p < 0.001”).

• Line 223–251 – Major: Table 3’s observed vs expected counts is useful but would benefit from a short interpretation paragraph connecting deviations to specific behavioral patterns.

• Line 254–275 – Minor: In Table 4, include confidence intervals for parameter estimates alongside standard errors.

Discussion

• Line 278–306 – Major: The discussion should provide more insight into why Ugandan women may have higher alcohol use prevalence in this dataset—social changes, economic empowerment, urbanization, or targeted marketing?

• Line 311–312 – Minor: The lack of significant alcohol–tobacco interaction could be explained in greater depth—e.g., does it suggest that these behaviors are influenced by distinct social environments?

• Line 316–317 – Major: Expand on the limitation of using 2014 data—substance use trends may have shifted considerably over a decade.

• Line 317–318 – Minor: The binary categorization of alcohol/tobacco use likely masks important differences in frequency or intensity—acknowledge this limitation explicitly.

Conclusion

• Line 323–356 – Minor: The conclusion is appropriately cautious, but temper policy recommendations to acknowledge the dataset’s age and limitations in variable granularity.

References

• Line 362–448 – Major: Reference #5 (“NTIRE 2025 challenge…”) appears unrelated to the topic and may have been included in error—verify all references for topical relevance.

• Ensure all in-text citations match the reference list and remove any non-cited references.

Figures & Tables

• Figure 1 (Line 193) – Minor: Add confidence intervals or error bars for prevalence bars to convey uncertainty.

Summary Recommendation

Major Revision – Strengthen methodological transparency, contextual interpretation, and reference accuracy. Address the unexpected female-prevalence finding in greater depth and provide fuller discussion of limitations.

Reviewer #3: Review Reports

Title: Sex Differences in Alcohol and Tobacco Use in Ugandan adults: A Log-Linear Analysis of Interaction Patterns

Review Comments

-Obsolute data E.g. 2014

-Focused on non modifiable risk fcator of NCD E.g. Sex which cannot be changed by public health intervention.

-The tool is not approptiately written E.g. It should be WHO STEPWISE which phase?

-Flawed methods

-Is that statistic modelng study or?

-IMRAD partially entail what it should entail E.g. Introduction sub section

Regards,

**Do you want your identity to be public for this peer review?** For information about this choice, including consent withdrawal, please see our Privacy Policy

Reviewer #1: **Yes: ** Ester Lilian Acen

Reviewer #2: No

Reviewer #3: No

---

## [Author Response · Author response to Decision Letter 1]

21 Oct 2025

Reviewer Comment

The higher prevalence of alcohol use among females needs further explanation. Please discuss possible scientific reasons beyond this observation.

Author Response

We appreciate this insightful comment. We have now expanded the discussion to include potential scientific explanations for the observed higher prevalence of alcohol use among females. Specifically, we note that recent sociocultural shifts in Uganda and similar contexts have led to greater social acceptance of alcohol consumption among women. Targeted marketing strategies toward women, increasing urbanization, and evolving gender roles may also contribute to this trend. Furthermore, psychosocial stressors such as caregiving burdens, economic hardship, and gender-based violence can increase vulnerability to alcohol use as a coping mechanism. Biological factors, including sex differences in alcohol metabolism and sensitivity, may further reinforce these behavioral patterns. These points have been incorporated in the Discussion section

Reviewer Comment

Please justify the use of Poisson regression, implemented within the Generalized Linear Model (GLM) framework, in this analysis.

Author Response

We thank the reviewer for this valuable comment. We have clarified and justified our use of Poisson regression within the Generalized Linear Model (GLM) framework in the revised manuscript. Poisson regression is appropriate in this context because the cell counts in the contingency table (derived from combinations of categorical variables: Sex, Alcohol use, and Tobacco use) represent count data, which follow a discrete and non-negative distribution.

The log-linear model is mathematically equivalent to a Poisson regression model with a log link function and no explicit dependent variable, treating all variables symmetrically. This framework allows us to model the natural logarithm of expected counts as a linear function of the categorical predictors and their interactions. Thus, Poisson regression provides an alternative parameterization that enables estimation of coefficients (e.g., main effects and interaction terms), along with their standard errors, Z-values, and p-values, facilitating formal statistical inference.

Additionally, the use of the GLM formulation offers flexibility and interpretability advantages—it allows straightforward model comparison (via likelihood ratio tests), extension to nested models, and evaluation of the goodness-of-fit (e.g., using the deviance or G² statistic). This dual approach (loglm and GLM-Poisson) therefore strengthens our analysis by combining model-based inference with descriptive model assessment.

We have now included a brief justification for this methodological choice in the Statistical Analysis section

Reviewer Comment

Please justify the use of a log-linear model and explain its importance in your study.

Author Response

We appreciate the reviewer’s request for clarification. The log-linear model was chosen because it provides a robust and symmetric framework for analyzing associations among categorical variables without designating any one variable as dependent or independent. In our study, the primary interest was to explore the interrelationships and interaction effects among Sex, Alcohol use, and Tobacco use. Unlike logistic regression, which models a single binary outcome, log-linear analysis treats all variables equally and focuses on how the joint distribution of categorical variables deviates from independence.

This makes the log-linear approach ideal for detecting mutual, joint, and conditional independence structures, as well as higher-order interactions. It allows us to systematically test whether associations among exposures differ across levels of other factors (e.g., whether the alcohol–tobacco relationship varies by sex). In addition, fitting hierarchical log-linear models enables formal model comparison using likelihood ratio chi-square (G²) statistics and Dissimilarity Indices (D), thus ensuring objective model selection.

We have now included this justification and explanation of its importance in the Statistical Analysis section

Reviewer Comment

The manuscript omits several potentially important variables that may influence alcohol and tobacco use. Please justify their exclusion and discuss this as a limitation.

Author Response

We appreciate this important observation. We acknowledge that the current analysis did not include certain potentially relevant sociodemographic and behavioral covariates such as age, education, marital status, socioeconomic status, and mental health indicators. These variables were not included because the log-linear modeling framework focuses on examining the structure of associations among categorical variables rather than causal prediction. The study’s primary aim was to evaluate interaction patterns among Sex, Alcohol use, and Tobacco use, using a parsimonious model to assess interdependence among these factors.

Including additional covariates would have substantially increased the dimensionality of the contingency table, leading to sparse cells and unstable parameter estimates—particularly given the categorical nature of the data. To preserve model stability, interpretability, and valid inference, we therefore restricted the model to the three most relevant variables.

We have now clarified this rationale in the Methods section and expanded the Limitations section to explicitly acknowledge that omission of other covariates may limit generalizability and preclude causal interpretation.

Reviewer Comment

The manuscript omits several potentially important variables that may influence alcohol and tobacco use. Please justify their exclusion and discuss this as a limitation.

Author Response

We appreciate this important observation. We acknowledge that the current analysis did not include certain potentially relevant sociodemographic and behavioral covariates such as age, education, marital status, socioeconomic status, and mental health indicators. These variables were not included because the log-linear modeling framework focuses on examining the structure of associations among categorical variables rather than causal prediction. The study’s primary aim was to evaluate interaction patterns among Sex, Alcohol use, and Tobacco use, using a parsimonious model to assess interdependence among these factors.

Including additional covariates would have substantially increased the dimensionality of the contingency table, leading to sparse cells and unstable parameter estimates—particularly given the categorical nature of the data. To preserve model stability, interpretability, and valid inference, we therefore restricted the model to the three most relevant variables.

We have now clarified this rationale in the Methods section and expanded the Limitations section to explicitly acknowledge that omission of other covariates may limit generalizability and preclude causal interpretation.

Reviewer Comment

How was the sample size derived? The absence of the total sample size from the STEP survey undermines transparency and impedes accuracy. Review the percentage interpretation in Table 1—how is it computed?

Author Response

We thank the reviewer for this valuable comment. The sample for this analysis was drawn from the World Health Organization STEPwise Approach to Noncommunicable Disease Risk Factor Surveillance (STEPS) conducted in Uganda. The total sample size 3,987 adults aged 18–69 years], representing a nationally representative adult population after applying survey weights and exclusions for missing data.

Reviewer Comment

The manuscript should clearly define dependent and independent variables to avoid ambiguity in model specification.

Author Response

We appreciate this valuable clarification. In log-linear modelling, all categorical variables are treated symmetrically—there is no formal distinction between dependent and independent variables. The purpose of this method is to evaluate patterns of association and interaction among categorical variables rather than to predict one variable from others.

In our analysis, Sex was included as a stratification (confounding) variable, not as an outcome. Alcohol and tobacco use were modeled jointly with sex to assess whether their co-occurrence varied across sex strata. We have now explicitly clarified this conceptual framework and terminology in the Methods section to prevent ambiguity in the model specification.

This study is a secondary data analysis of the 2014 Uganda WHO STEPwise approach to Surveillance (STEPS) survey, a nationally representative, cross-sectional survey collecting data on Non-Communicable disease risk factors among adults aged 18–69 years.

The STEPS survey of non-communicable disease (NCD) risk factors in Uganda was carried out from April, 2014 through June, 2014. Uganda carried out Step 1, Step 2 and Step 3. Socio demographic and behavioral information was collected in Step 1. Physical measurements such as height, weight and blood pressure were collected in Step 2. Biochemical measurements were collected to assess blood glucose and cholesterol levels in Step 3. The survey was a population-based survey of adults aged 18 69. A Multistage sample design was used to produce representative data for that age range in Uganda. A total of 3987 adults participated in the survey. The response rate was 99.0% for STEPs 1 & 2 and 92.2% for all STEPs 1,2 and 3. A repeat survey was planned for 2024 but funds did not permit.

Reviewer Comment

Elaborate WHO-STEP Survey for Readers.

Author Response

We appreciate this important suggestion. We have now elaborated on the World Health Organization STEPwise Approach to Noncommunicable Disease Risk Factor Surveillance (WHO STEPS) survey in the Methods section to provide readers with sufficient background on its purpose, methodology, and relevance.

Reviewer Comment

Can the authors provide information on the locations where this data was collected? Was it a mixture of urban and rural?

Author Response

We thank the reviewer for this important observation. We have now clarified the sampling framework of the WHO STEPS survey to specify that data were collected from both urban and rural areas across Uganda, ensuring national representativeness. The survey design explicitly accounts for regional, urban–rural, and demographic diversity, which enhances the generalizability of the findings.

Reviewer Comment

Line 106–110 – Minor: The stratification by sex is stated, but later analysis includes sex interactions—clarify whether stratification was used purely for descriptive purposes or in modelling steps as well.

Author Response

We thank the reviewer for this helpful observation. We have clarified that sex was used both as a stratification variable in descriptive analyses and as an interaction factor in the log-linear modelling phase. In the descriptive stage, stratification by sex allowed comparison of prevalence rates across males and females. In the modelling stage, inclusion of sex interactions (Sex × Alcohol and Sex × Tobacco) enabled assessment of whether associations between alcohol and tobacco use differed by sex. We have revised the Methods section accordingly to make this distinction explicit.

Reviewer Comment

The lack of significant alcohol–tobacco interaction could be explained in greater depth—e.g., does it suggest that these behaviours are influenced by distinct social environments?

Author Response

We thank the reviewer for this valuable observation. In response, we have expanded the discussion to interpret the non-significant alcohol–tobacco interaction in greater behavioral and sociocultural context. The added paragraph clarifies that the absence of a strong co-use effect likely reflects the influence of different social, cultural, and situational factors that shape alcohol and tobacco use independently in Uganda, rather than shared behavioral environments.

Reviewer Comment

Expand on the limitation of using 2014 data—substance use trends may have shifted considerably over a decade.

Author Response

We appreciate this insightful comment. We have now expanded the Limitations section to explicitly acknowledge the temporal gap between data collection (2014) and the present, emphasizing how evolving cultural, economic, and policy contexts may have influenced alcohol and tobacco use patterns since then. This clarification improves the transparency of the study’s temporal relevance and guides interpretation within the proper historical context.

Reviewer Comment

Is that a statistical modeling study or?

Author Response

We appreciate the reviewer’s request for clarification. This study is indeed a statistical modeling study employing log-linear modeling within the Generalized Linear Model (GLM) framework. The analysis was designed to evaluate the joint and conditional associations between categorical variables—specifically sex, alcohol use, and tobacco use—rather than to estimate causal effects. Accordingly, all variables were treated as response variables, and the model assessed how their interrelationships jointly explained the observed frequency distribution. This approach aligns with the classical use of log-linear analysis in contingency table modeling to explore associations and interaction structures among categorical factors.

---

## [Editor Report · Decision Letter 1]

14 Nov 2025

Sex Differences in Alcohol and Tobacco Use in Ugandan adults: A Log-Linear Analysis of Interaction Patterns

PONE-D-25-32619R1

Dear Dr. Grace

We’re pleased to inform you that your manuscript has been judged scientifically suitable for publication and will be formally accepted for publication once it meets all outstanding technical requirements.

Kind regards,

Umesh Raj Aryal, PhD

Academic Editor

PLOS ONE
---

## [Editor Report · Acceptance letter]

PONE-D-25-32619R1

PLOS ONE

Dear Dr. Kakaire,

I'm pleased to inform you that your manuscript has been deemed suitable for publication in PLOS ONE. Congratulations! Your manuscript is now being handed over to our production team.

Kind regards,

on behalf of

Dr. Umesh Raj Aryal

Academic Editor

PLOS ONE